# Distinct molecular cues ensure a robust microtubule-dependent nuclear positioning in the *Drosophila* oocyte

Nicolas Tissot[1], Jean-Antoine Lepesant[1], Fred Bernard[1], Kevin Legent[1], Floris Bosveld[2], Charlotte Martin[2], Orestis Faklaris[1], Yohanns Bellaïche[2], Maïté Coppey[1] & Antoine Guichet[1]

Controlling nucleus localization is crucial for a variety of cellular functions. In the *Drosophila* oocyte, nuclear asymmetric positioning is essential for the reorganization of the microtubule (MT) network that controls the polarized transport of axis determinants. A combination of quantitative three-dimensional live imaging and laser ablation-mediated force analysis reveal that nuclear positioning is ensured with an unexpected level of robustness. We show that the nucleus is pushed to the oocyte antero-dorsal cortex by MTs and that its migration can proceed through distinct tracks. Centrosome-associated MTs favour one migratory route. In addition, the MT-associated protein Mud/NuMA that is asymmetrically localized in an Asp-dependent manner at the nuclear envelope hemisphere where MT nucleation is higher promotes a separate route. Our results demonstrate that centrosomes do not provide an obligatory driving force for nuclear movement, but together with Mud, contribute to the mechanisms that ensure the robustness of asymmetric nuclear positioning.

[1] Polarity and Morphogenesis Team, Institut Jacques Monod, CNRS UMR 7592, Université Paris-Diderot, Sorbonne sParis Cité, Paris Cedex 75205, France. [2] Polarity, Division and Morphogenesis Team, Institut Curie, CNRS UMR 3215, INSERM U934, 26 rue d'Ulm, Paris Cedex 05 75248, France. Correspondence and requests for materials should be addressed to A.G. (email: antoine.guichet@ijm.fr).

The movement and the positioning of the nucleus play critical roles in many cellular and developmental contexts, including cell division, cell polarity and motility as well as tumour progression[1]. The cytoskeleton plays a central function in nuclear positioning. Actin filaments, microtubules (MTs) as well as associated motor proteins are instrumental in this process. In many cases, MTs participate in the precise localization of the nucleus in close association with a centrosome acting as MT organizing centre and the MT-associated motors Dynein and Kinesin l (ref. 2). This is illustrated by their roles in interkinetic nuclear migration in developing neuronal progenitors, or bidirectional nuclear movement during hypoderm development in the *Caenorhabditis elegans* embryo[3,4]. Alternatively, MTs can also function independently of the centrosomes, as it is the case for nucleus centration and nucleus anchoring along the cell cortex in vertebrate developing muscles[5], or for nuclear positioning in fission yeast[6,7]. The molecular mechanisms that drive such centrosome-independent nuclear migration are less well understood. In the *Drosophila* oocyte, the positioning of the nucleus is a key event for the organization of the MT-based polarized transport, which controls the asymmetric localization of mRNAs that encode determinants of the polarity axes of the future embryo[8–11]. There is a dual relationship between the nucleus and the MTs. The positioning of the nucleus influences the organization of the MT network in the oocyte[11–13], while reciprocally MTs themselves are also instrumental in nuclear migration[12,14–16]. From stages 6 to 7 of oogenesis, the oocyte nucleus migrates from the posterior to the anterior side of the oocyte and is subsequently anchored at the boundary between the plasma membrane of the anterior margin and the lateral membrane. The molecular mechanisms underlying these processes are not yet fully understood. Interestingly, the anchorage of the nucleus at the anterior can occur randomly at any position along the circumference of the lateral cortex of the oocyte[17]. This asymmetric nuclear positioning specifies the identity of the antero-dorsal cortex and initiates the establishment of dorsal-ventral polarity of the egg chamber and the future embryo[8,9,11,16]. While MTs are essential for all aspects of nuclear positioning, so far their associated motors have been shown to be only required for its correct anchorage[10,13,16,18–20].

Our current understanding of the mechanisms underlying the positioning of the oocyte nucleus was gained from the analysis of fixed tissues or two-dimensional live imaging[8,9,11,16,21]. However, the oocyte is a large cell in which migration of the nucleus proceeds in three dimensions within its entire volume. Using three-dimensional (3D) live imaging combined with photo-manipulation and genetics, we have undertaken a functional study of the forces and molecular mechanisms that drive the migration of the oocyte nucleus. We reveal that the dual contributions from the centrosomes and the NuMA homologue Mud at the nuclear envelope in a microcephaly protein Abnormal spindle-dependent mode, ensure the robustness of asymmetric nuclear positioning through alternative migratory routes.

## Results

**3D visualization of nuclear migration in the oocyte.** To investigate the mechanisms controlling the migration of the nucleus, we developed a live-cell imaging assay using dissected egg chambers cultured in a micro chamber. Decoration of the nuclear envelope was achieved with a tagged β–importin (Fs(2)Ket-GFP), which interacts and co-localizes with nucleoporins[22], while the plasma membrane was labelled by the PIP2-binding domain of the phospholipase- δ1 tagged with RFP (PH[PLC-δ1]-RFP)[23,24] (Fig. 1a,b). The precise positioning of the nucleus with regard to the oocyte plasma membrane was

determined at each time point from a still of 40 optical sections along the z axis, taken at 15 min intervals, for up to 10 h on a spinning disc confocal microscope. This allowed the complete process of nuclear migration to be visualized in a large number of samples in 3D (Fig. 1c). Importantly, shorter movies with time points separated by 1 min only, provided similar results indicating that a 15 min frame-rate is sufficient to record the complexity of the nuclear migration. Before its migration the nucleus does not stand still (Fig. 1c) but appears to be in constant motion around a central position (Supplementary Fig. 1A), suggesting that migration may be initiated by an imbalance in opposing forces. The migrating nucleus then follows a 3D trajectory that ends with its nesting and anchorage at the intersection between the anterior and the lateral plasma membranes (Fig. 1c).

**Relation between directionality and indentation.** In many organisms and cell types, the centrosomes are associated with nuclear migration[1]. Recently, a two-dimensional live imaging analysis has shown that, in the *Drosophila* oocyte, centrosomes drive nuclear mobility, through the nucleation of MTs that can exert a pushing force on the nucleus, and indent its envelope[16]. However, we frequently observed that the deformation of the nuclear envelope was not aligned with the actual displacement of the migrating nucleus (Fig. 1d; Supplementary Movie 1). Furthermore, cases with several or only very weak deformations of the nuclear envelope were also observed. This led us to reinvestigate the correlation between the nuclear indentations and the direction of nucleus migration. The vector resulting from the nuclear deformations (Supplementary Fig. 1B) was measured for each time point ($n = 429$) in 50 independent migration recordings, and the angle between this vector and the one corresponding to the observed nuclear displacement was measured (see Methods; Supplementary Fig. 1C). In 52% of the cases ($n = 223$), the angle was greater than 90° (Fig. 1e). This indicated that in at least half of the cases, there was no correlation between the indentation and the directionality of the nucleus motion. Nuclear migration is a rather slow process, it is thus possible that the net displacement of the nucleus only matched cases where this angle was smaller than 90°. If so, angles smaller than 90° should correlate with increase net displacement. This possibility was thus investigated by evaluating the cross-correlations between the distance covered by the nucleus and the angle between indentation force and displacement vector, with a lag of 0, 15, 30, 45, 60 and 75 min. For each migration, the maximum distance covered by the nucleus between two time points is normalized to 1 (Supplementary Fig. 1D). We observed no correlation between the normalized net displacement and the angles as revealed by a correlation coefficient $R^2$ of 0.016 (Supplementary Fig. 1D). We also considered the possibility of a delay for the correlation between the displacement and the angle between the indentation force and the displacement vector. We evaluated the cross-correlation considering a potential lag time up to 75 min (Fig. 1f). Again, we observed no correlation as revealed by the various correlation coefficients. Together, these results indicated that there was no correlation between the directionality of the nuclear migration and the forces creating nuclear indentations. We, therefore, conclude that although centrosome-dependent MT pushing forces contribute to nuclear positioning[16], those pushing forces applied onto the nuclear envelope are unlikely to be solely responsible for nuclear displacement. Our findings reveal that nuclear migration is a more complex process than previously anticipated. This prompted us to assess in detail the nature of the forces at work during nucleus movements.

**MT forces applied on the nucleus.** Previous studies have shown that the oocyte nucleus fails to migrate when MTs are

depolymerized by colcemid (Supplementary Movie 2)[12,14–16]. Since MT growth or shrinkage could generate either pulling or pushing forces, respectively, we wondered whether MTs control nuclear mobility by exerting such forces on the nucleus (Supplementary Fig. 2A). To discriminate between these possibilities, we first analysed the MT distribution during nuclear migration. We reasoned that if the nucleus is mostly pulled, relevant MTs should be located between the future antero-dorsal corner and the nucleus during migration. Conversely, if the nucleus is pushed, the bulk of relevant MTs should be located behind the nucleus during migration. MT distribution was monitored using the MT-associated protein

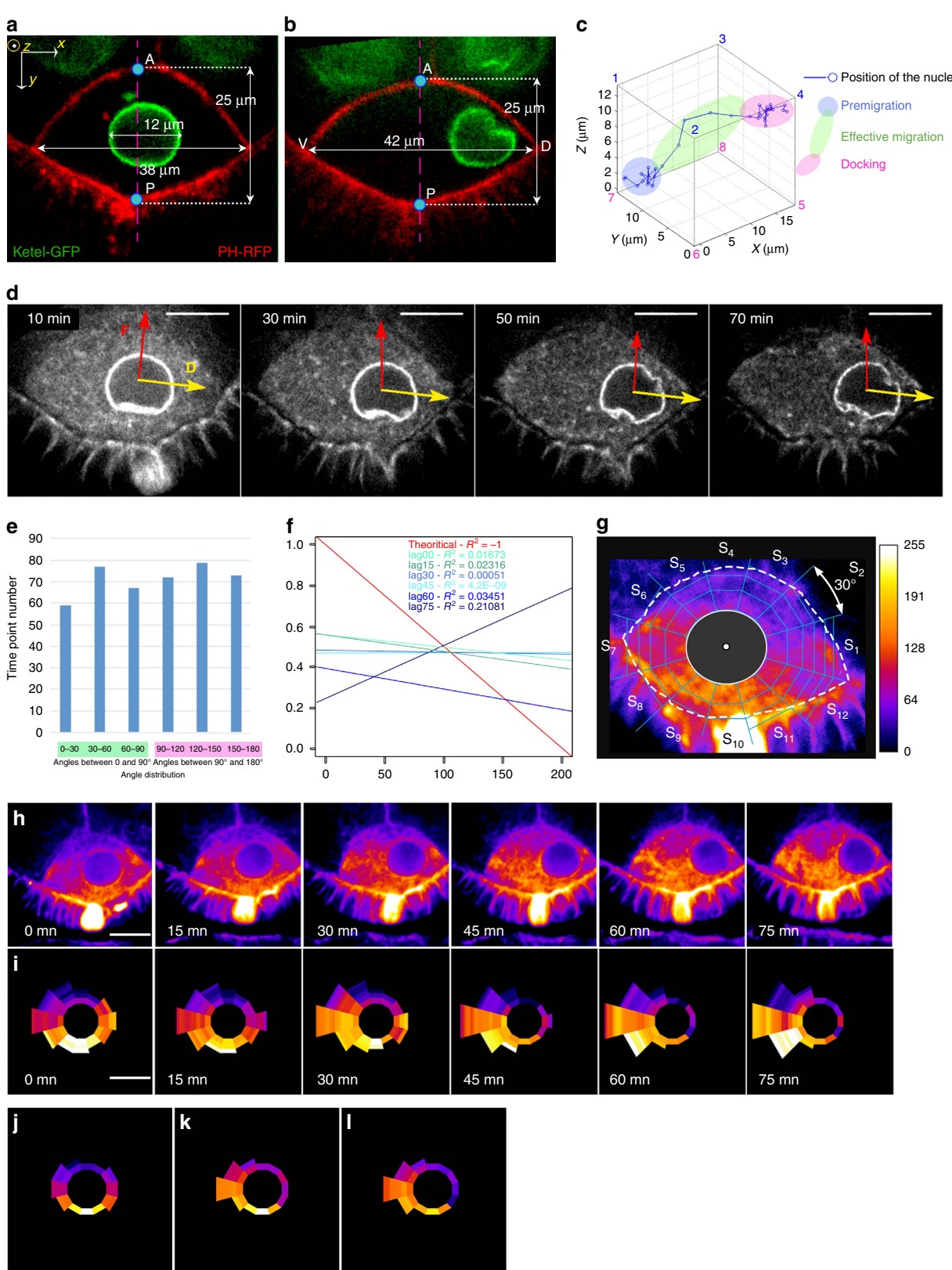

(MAP) Jupiter fused to GFP, which allowed live imaging of the MTs in the developing oocyte[25]. Before nuclear migration most of the MTs were concentrated between the nucleus and the posterior cortex (Fig. 1g). Jupiter-GFP distribution based on fluorescence intensity measurement during nuclear migration (Fig. 1g) was analysed in 31 independent events. In all cases, the bulk of the signal was concentrated posteriorly to the nucleus during its migration (Fig. 1h–l; Supplementary Movie 3). This result suggested that the nucleus is rather pushed than pulled by MTs. To further investigate this point, a pulsed laser-based nano-ablation set-up was used to locally perturb the MT network to probe the forces applied onto the nucleus. We first analysed the immediate effect of nano-ablation with a ultraviolet picosecond laser on short-term nuclear displacement. Ablations between the anterior cortex and the nucleus either shifted the nucleus anteriorly or posteriorly or did not affect its position (Supplementary Fig. 2B–E,J; Supplementary Movie 3). Conversely, ablations between the nucleus and the posterior cortex shifted the nucleus only posteriorly (Supplementary Fig. 2F–J; Supplementary Movie 4), suggesting that a pushing force holds in place the nucleus at that time. We next investigated the long-term consequences on the nuclear migration of nano-ablation with a two photon-mediated femtosecond laser. It should be noted that in this case, MT regrowth after photo-ablation would occur on a much shorter timescale than the studied effects on nuclear positioning. When applied all around the nucleus before migration onset, nano-ablation prevented subsequent targeting to the antero-dorsal cortex (Fig. 2a,b; Supplementary Movie 6), confirming that forces applied to the nucleus participate to its displacement. Importantly, we also found that targeting two photon-mediated pulses directly onto the nucleus did not affect its eventual migration (Supplementary Movie 7). When nano-ablation was performed between the nucleus and the anterior cortex, migration of the nucleus proceeded successfully in a majority of cases and was impaired in 19.3% of the cases only ($n = 31$) (Fig. 2c,d,g; Supplementary Movie 9). On the contrary, when ablation was applied between the nucleus and the posterior margin of the oocyte, migration of the nucleus was impaired in a large majority of the cases (58.3%, $n = 36$) (Fig. 2e–g; Supplementary Fig. 2K,L; Supplementary Movie 8). Thus, these nano-ablation experiments further indicate that a set of forces located behind the nucleus is responsible for its motility. Taken together with the analysis of MT distribution, our results suggest that the nucleus is rather pushed than pulled by the MTs.

**The nucleus migrates through alternative routes.** Having established that the nucleus is pushed, we aimed to gain insight into the mechanisms involved in the nuclear thrust. To this end,

we performed a quantitative analysis of 24 migrations in 3D (see methods and Supplementary Fig. 3), and observed that the targeting of the nucleus to the antero-dorsal cortex is a variable process that can be achieved through different paths. In the vast majority of cases, a biphasic migration pattern was observed. The nucleus hit either the anterior plasma membrane (APM) or the lateral plasma membrane (LPM) before riding along it to reach its final destination (Fig. 3a,c,e; Supplementary Movies 10,11). In contrast, in 2 cases only (8%), the nucleus migrated straight to the antero-dorsal cortex (STAD) and did not contact any plasma membrane before its arrival (Fig. 3b,e; Supplementary Movie 12). Nuclear velocity along the APM or LPM routes was similar ($0.14 \pm 0.02\,\mu m\,min^{-1}$). For a precise comparison of these events in 3D, the oocyte was assimilated to a geometrical model structure with an axial symmetry along the antero-posterior (AP) axis (see methods and Supplementary Fig. 3A). For standardization, each contact point along the oocyte plasma membrane was positioned onto a normalized diagram representing a reference oocyte (see methods, Supplementary Fig. 3A–F). The number of contacts on either the APM or the LPM was not significantly different ($P$ value $= 0.683 > > 0.05$) (Fig. 3d,e) indicating that the number of nuclei migrating along the APM or the LPM was similar. However, the distribution of the initial contact points along the APM and the LPM was statistically different. The APM contacts showed an even distribution spanning the entire length of the cortex, whereas contacts along the LPM were excluded from the lateral-most region (Fig. 3d). The difference of initial contact point repartition between APM and LPM could be due to biases imposed by the oocyte organization or geometry. Alternatively, this difference could also suggest that nuclear migration can proceed through alternative tracks (Fig. 3f), which may involve specific molecular regulators that impinge on the trajectories.

**Centrosomes favour nuclear displacement along the APM.** Our results suggested that the centrosome does not provide the only force driving nuclear displacement (Fig. 1). Nevertheless, its possible contribution to the commitment of the nucleus in alternative migratory routes remained to be assessed. During early oogenesis, the centrosomes of the 15 nurse cells migrate through the ring canals into the oocyte, shortly after its determination[26], forming a potential group of 16 centrosomes. We noticed that in 92% of the cases, centrosomes, revealed by the centriolar marker Asterless (Asl) fused to tdTomato[27], coalesced in a compact structure that migrated in close association with the nucleus (Fig. 4a,b; Supplementary Movie 13). In 8% of the oocytes the centrosomes displayed a more scattered pattern in the vicinity of the nucleus (Fig. 4c,d; Supplementary Movie 14)

**Figure 1 | Nuclear indentation does not correlate with the directionality of the migration.** (**a**,**b**) Fs(2)Ket-GFP (green) outlining nuclear envelope and PH[PLC-∂1]-RFP (red) revealing plasma membrane were used to monitor nucleus migration between stages 6 (**a**) and 7 (**b**). Oocyte orientation and nuclear dimensions are indicated (A: Anterior, P: Posterior, D: Dorsal and V: Ventral). The pink A-P dash line (**b**) highlights the symmetry axis of the oocyte. (**c**) 3D rendering of a discrete nuclear migration event from the posterior to the antero-dorsal cortex. The position of the nucleus centre enables the identification of three consecutive phases: premigration (blue), migration-proper (green) and cortical docking (pink). (**d**–**f**) Correlation between indentation and directionality. Angles between the vector **F** of the forces (red in **d**) and the vector **D** of nuclear displacement (yellow in **d**) were calculated at each time point of time-lapse movies. Example of a time-lapse recordings (**d**) used for this calculation. (**e**) Histogram of the distribution of angles for the 223 time points analysed from 50 independent nuclear displacements. (**f**) Correlation analysis between a potential phase lag and the correlation between the nuclear indentation and the nuclear displacement. The theoretical function of a perfect correlation is represented in red with an $R^2$ of $-1$. Different lag-times were considered: 0, 15, 30, 45, 60, 75 min (see Methods). (**g**–**i**) MT densities were monitored over different time points and oocytes. Trapezoid boxes of a fixed volume were defined (**g**) by subdividing the oocyte cytoplasm into 12 sectors (S1–S12) surrounding the nucleus that were further subdivided into trapezoids. A colour-coded heat map spanning 256 levels of fluorescence highlights fluorescence intensities (**g**). Dashed and solid lines highlight oocyte and nucleus limits. (**h**,**i**) Selected frames at indicated time points from a representative example of a time-lapse movie showing MT densities (**h**) and the graphical representation (**i**) of the same images. Scale bar, 10 μm (**d**,**h**,**i**). (**j**–**l**) Mean MT densities for 31 nuclear migrations analysed at three time points: prior to migration (**j**), mid-migration (**k**) and migration completion (**l**).

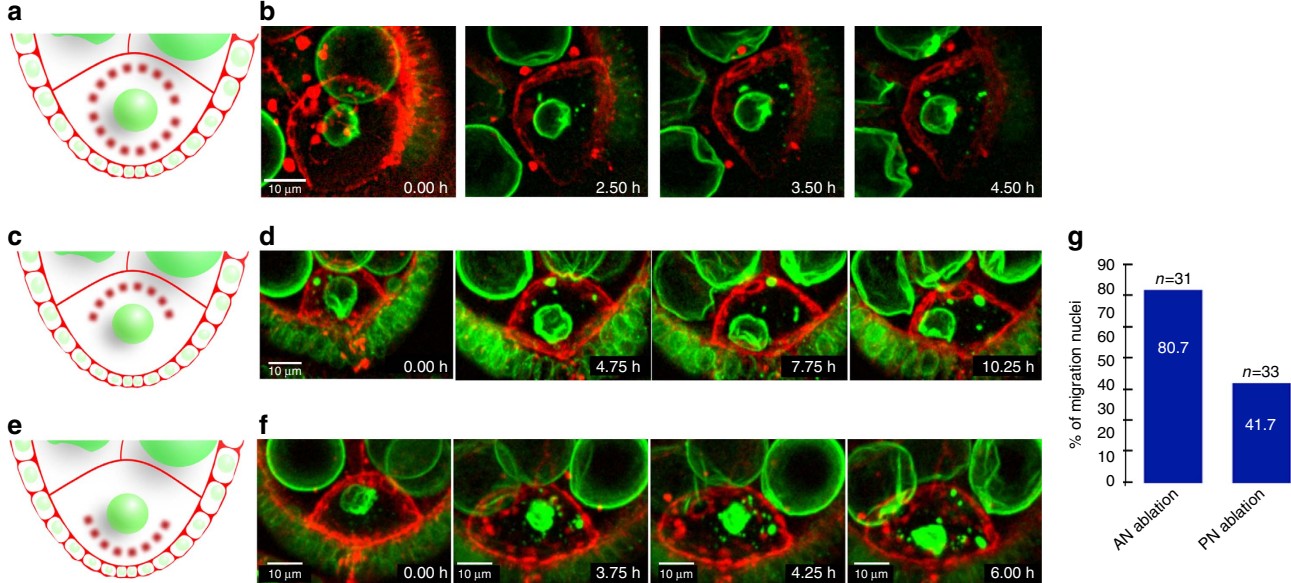

**Figure 2 | Nano-ablation-mediated analysis of forces applied onto the nucleus during its migration.** Laser ablation experiments were performed on oocytes expressing Fs(2)Ket-GFP to label nuclei (green) and PH$^{PLC-\partial1}$-RFP to label plasma membranes (red). (**a,c,e**) Schematic representations of the photo-ablated areas (red dots) before nuclear migration. (**b,d,f**) Selected frames extracted from time-lapse movies at the indicated time after ablation. (**b**) Representative example of a nucleus that does not migrate after ablation all around the nucleus (**d**) Representative example of a nucleus migration after nano-ablation performed anteriorly to the nucleus. (**f**) Representative example showing a nucleus that does not migrate after ablation performed posteriorly to the nucleus. (**g**) Bar plots presenting the percentage of oocytes in which nuclear migration is effective after anterior ($n = 31$) and posterior ($n = 33$) nano-ablations.

($n = 75$). Similar results were observed on fixed tissues and with other centrosomal markers such as Pericentrin-like protein. This raised the question of whether all scattered centrosomes or only those close to the nucleus were active. 3D-structured illumination microscopy showed that two different components of the essential pericentriolar material, Spindle defective 2 (Spd-2) and Centrosomin (Cnn)[28], surrounded all centrioles, as revealed by the co-localization of Asl-tdTomato with Spd-2-GFP ($n = 6$) (Fig. 4e,f) and Cnn-GFP ($n = 10$) (Fig. 4g,h). This suggested that all centrosomes were active[29], even when they were not in close vicinity to the nucleus. We thus investigated the potential involvement of centrosomes in the mechanisms that commit the nucleus in alternative migratory routes. Centrosomes were depleted from the oocyte by knocking down the *sas-4* or *asl* transcripts by RNAi[30] and the RNAi efficiency was verified (see methods). This resulted in a decrease of the nuclear velocity ($0.10 \pm 0.01 \, \mu m \, mn^{-1}$) compared to wild type (Supplementary Movie 15). In addition, out of 23 recordings, we observed 48 % of nuclear migrations along the LPM, 20% along the APM and 30% directly through the cytoplasm (STAD) (Fig. 4i). This shift in the distribution of migratory routes indicated that centrosomes do contribute to the migration of the nucleus and that they favour a displacement along the APM (Fig. 3f).

**MTs are asymmetrically nucleated at the nuclear envelope.** Our results indicated that centrosomes favour a specific trajectory (APM) and that they do not provide the only force that drives nuclear migration in agreement with our findings on nucleus indentation. We thus decided to investigate whether other MT nucleation sites may contribute to nucleus displacement and would involve alternative molecular players. We specifically focused on the nuclear envelope since MTs nucleation had been previously shown to occur in its vicinity[12].

MT organization within the oocyte is complex and the detection of MT nucleation sites is challenging[15]. To improve the ability to locate MT nucleation centres and record MT dynamics, we developed a live-imaging assay for the repolymerization of Jupiter-GFP decorated MTs. MT regrowth was recorded after colcemid-induced depolymerization, followed by drug inactivation by UV laser illumination[31], in a specific region of interest (Fig. 5a–h; Supplementary Movie 16). In a similar assay, the MT plus-end binding protein EB1-GFP[32] allowed the directionality of MT regrowth to be monitored more precisely (Fig. 5i–l; Supplementary Movie 17). Jupiter-GFP or EB1-GFP-positive MT nucleation centres were observed at multiple sites and at different locations within the oocyte, most of them being clearly distinct from the centrosomes, as verified by co-labelling with Asl-tdTomato (Supplementary Fig. 2M). It was noticed in particular that centrosome-independent MT nucleation was significant in the vicinity of the nuclear envelope (Fig. 5i–l). Investigation of the 3D distribution of MT nucleation sites around the nucleus (Fig. 5m–p) showed that 90% of the nuclei ($n = 23$) had more MT nucleation sites (2.1-fold in average) in their hemisphere facing the posterior of the oocyte (Fig. 5p). Interestingly, the distribution of MT nucleation sites on this hemisphere correlated with the MT enrichment observed behind the nucleus during migration (Fig. 1f–l) as well as with the asymmetric perinuclear distribution of $\gamma$-tubulin[12,15]. These observations support a model in which extending MTs nucleated at the posterior side of the nuclear envelope exert a pushing force that contributes to the nuclear displacement.

**Mud promotes nuclear displacement along the LPM.** During oogenesis in *C. elegans*, the MAP NuMA is essential for controlling both the precise positioning of the meiotic spindle and the focusing of the MT minus ends[33]. Interestingly, we and others[16] found that the protein Mud, its fly ortholog, shows an

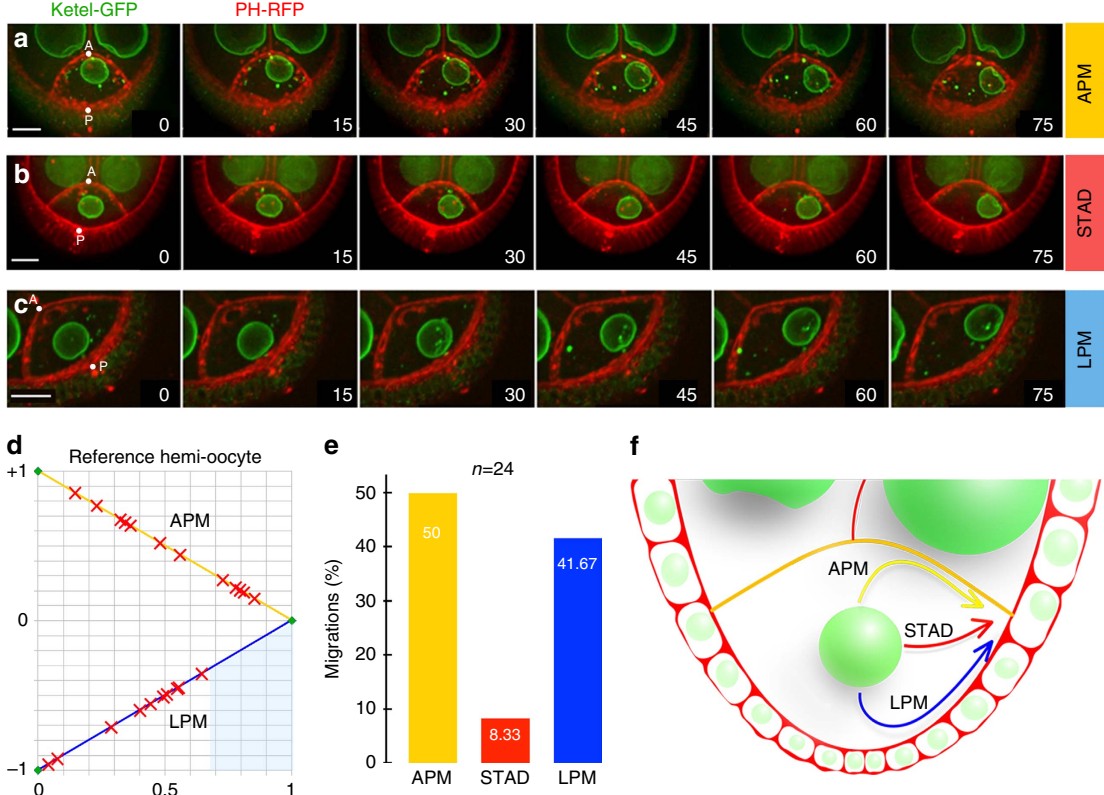

**Figure 3 | The nucleus migrates to the antero-dorsal cortex through alternative migratory routes. (a–c)** Selected frames extracted from time-lapse movies illustrating the three distinct migratory paths: along the APM (**a**), through the cytoplasm (STAD) (**b**), or along the LPM (**c**). Nuclei (green) and plasma membranes (red) are labelled by Fs(2)Ket-GFP and PH$^{PLC-\partial 1}$-RFP respectively. Time (min) is indicated at the lower right corner. A: anterior, P: posterior. Scale bar, 10 μm. (**d**) Normalized diagram of a reference oocyte, where the yellow and blue lines represent the APM and the LPM respectively. For 22 independent migrations, initial contact points between the nucleus and the membranes are indicated with red crosses. (For more details see Supplementary Fig. 3). Two $\chi^2$-tests for the statistical analysis of contact point distribution were performed, with the null hypothesis $H_0$ of an even distribution along either the APM or the LPM. $P$ values are 77.9% ($>>5\%$) and 2.5% ($<5\%$) respectively. Hence, the distribution along APM is even, whereas the LPM initial contact points fall between abscises [0; 0.67] and are excluded from abscises [0.67; 1] (blue region). (**e**) Bar plots of the distribution of the three different migration paths taken by the nuclei. (**f**) Schematic illustration of the three nuclear migratory tracks in the oocyte.

asymmetric distribution around the nuclear envelope, specifically in the oocyte (Fig. 6a–c; Supplementary Movie 18), with increased levels on the nuclear-posterior hemisphere (Fig. 6g–i) where the number of MT nucleation sites is highest (Fig. 5m–p). *In vivo* SIM microscopy revealed that Mud was closely associated with the nuclear envelope (Fig. 6d–f). Furthermore, the asymmetric distribution of Mud along the nuclear envelope was unchanged in the absence of centrosomes ($n = 20$) (Fig. 7a). In addition, we found that the microcephaly protein Asp (Abnormal spindle), that interacts with Mud/NuMa[33] and is required for nuclear migration in the *Drosophila* neuroepithelium[34] is asymmetrically distributed around the nuclear envelope of the *Drosophila* oocyte similarly to Mud (Fig. 6g–l). We next investigated whether Asp controls Mud localization. Asp was depleted by RNAi-mediated knockdown from the oocyte and the efficiency was verified (see methods). Mud remained perinuclear although its asymmetry was lost (Fig. 6m,n). This result was further confirmed by antibody staining of Asp$^{E3}$/Asp$^{t25}$ mutant egg chambers (Fig. 6o). Hence, Asp is not necessary for mud localization at the nuclear envelope but for its asymmetric distribution.

Next, we assessed whether Mud controls nuclear migration. Mud was depleted from the oocyte by RNAi-mediated knockdown and its efficiency was verified (Methods). While nuclear migration velocity was mildly reduced upon RNAi-mediated knockdown of Mud ($0.10 \pm 0.02 \, \mu m \, min^{-1}$) (Supplementary

Movie 19), a substantial shift in the 3D repartition of the migratory routes was observed as compared to wild-type oocytes (Fig. 7e to be compared to Fig. 3e). Out of 23 recordings, we observed 73.9% of nuclear migrations along the APM, 13.05% along the LPM and 13.05% directly through the cytoplasm (STAD) (Fig. 7e). This indicates that Mud contributes to nuclear migration and specifically favours a displacement along the LPM. It is interesting to note that the distribution of the nuclear routes in the absence of Mud is complementary to the one observed in the absence of centrosomes (Fig. 4i). Importantly, upon Asp RNAi-mediated knockdown, we also observed a similar shift in the 3D repartition of the migratory routes as compared to wild-type oocytes. Out of 21 recordings, we observed 57% of nuclear migrations along the APM, 19% along the LPM and 24% directly through the cytoplasm (STAD) (Fig. 7d). Hence, taken together these results suggest that distinct molecular cues ensure nuclear positioning through alternative migratory routes.

**Mud and the centrosomes confer robustness to nuclear migration.** In the absence of either Mud or the centrosomes, the nucleus was still correctly positioned at later stages of oogenesis (Fig. 7c)[30]. These data suggested that, while Mud and the centrosomes favour distinct migratory routes, they might ensure proper nuclear positioning redundantly. This prompted us to investigate the

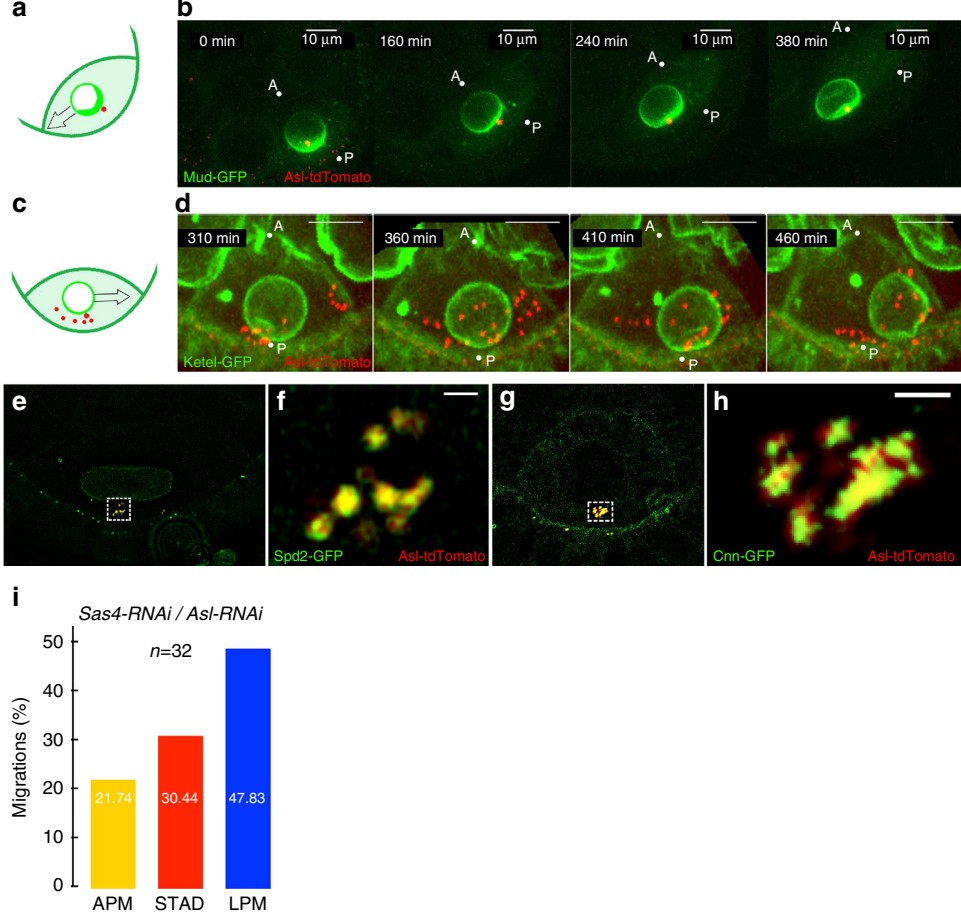

**Figure 4 | Centrosomes favour nuclear displacement along the APM.** During nuclear migration, centrosomes, revealed by Asl-tdTomato (red in **b,d**), coalesce in a compact structure (**a,b**) posterior to the nucleus revealed either by Mud-GFP (green in A) or Fs(2)Ket-GFP (green in **c**), but can be occasionally found scattered within the oocyte (**c,d**). Schematic representations (**a,c**). Selected frames extracted from representative examples time-lapse movies at the indicated times (**b,d**). (**e–h**) 3D-SIM imaging of the centrosomes. Close-ups of the boxed regions are displayed in **f,h**. (**e,f**) Spd-2-GFP (green) labels the PCM and Asl-tdTomato (red) labels the centrioles. The nucleus is highlighted by Lamin-GFP (green). (**g,h**) Cnn-GFP (green) labels the PCM and Asl-tdTomato (red) labels the centrioles. Scale bar, 500 nm. (**i**) Percentages of the various paths taken by nuclei during their migration when centrosomal activity is impaired. 3D-SIM, 3D-structured illumination microscopy. PCM, pericentriolar material.

effect of inactivating both *mud* and *Sas4*. In this condition, nuclear migration was abolished in 46% of the oocytes. In the remaining cases, migration along the LPM was lost (Fig. 7f). Consistently, in oocytes of later stages that did not degenerate, the nucleus was mispositioned in 35% of the cases ($n = 201$, Fig. 7g–i,). Taken together, these results indicate that in the *Drosophila* oocyte, contributions from the centrosomes and Mud at the nuclear envelope ensure the robustness of asymmetric positioning of the nucleus, through alternative migratory routes.

## Discussion

Our work reports that accurate cortical positioning is ensured redundantly by the contributions from the centrosomes and the MAPs Mud and Asp that favour alternative migratory trajectories. This unexpected level of robustness illustrates that the nuclear migration involves a regulated rather than a stochastic process, and most likely reflects the essential requirement for nuclear positioning in the establishment of the polarity axes of the oocyte and future *Drosophila*. Consistently, and as previously reported[16], we found that centrosomes often concentrated on the posterior side of the nucleus relative to the direction of its movement, and contribute to nuclear displacement. However,

they are dispensable for the nucleus to complete its migration. We also found that MTs are nucleated, independently of the centrosomes, at the level of the nuclear envelope and preferentially on its hemisphere facing the posterior cortex of the oocyte. A similar MT nucleation activity at the nuclear envelope has been observed in plant cells[35], as well as in differentiating muscles[5,36].

In an attempt to further decipher the forces involved in nucleus displacement, our laser ablation experiments showed that major forces involved are concentrated at the posterior of the oocyte and we interpreted this observation as an evidence of the nucleus being pushed, in accordance with a previous study[16]. However, further analyses of MT organization in *Drosophila* oocyte and especially the involvement of Mud and Asp, two previously shown MT minus-end associated proteins[37,38], make the interpretation of these photomanipulation experiments more complex. Indeed, this study reveals that the posterior part of the oocyte nuclear envelope could be hit by MTs nucleated from the active centrosomes as well as it could be a site of MT nucleation, indicating that MT plus-ends as well as MT minus-ends could be present at the same location. Therefore, further analyses of MT organization, and especially of molecular motors associated with MT, are required to fully understand how forces apply to the

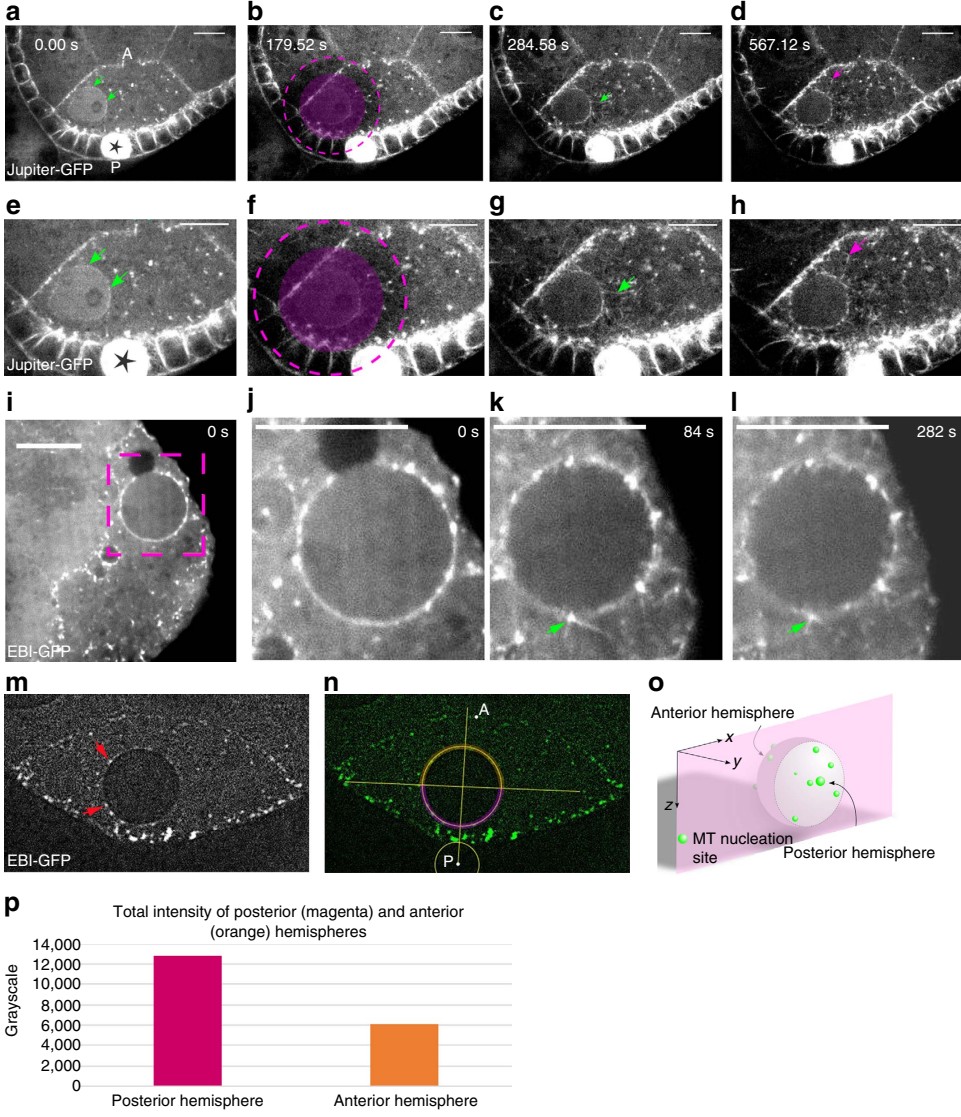

**Figure 5 | MTs are nucleated asymmetrically around the nuclear envelope.** (**a**–**h**) Selected frames extracted from a time-lapse movie illustrating Jupiter-GFP labelled MT nucleation after depolymerization by colcemid (**a**,**e**), and different time after local UV-mediated colcemid inactivation (**b**–**d**,**f**–**h**). The selected ROI for ultraviolet-pulse is marked by a solid magenta disc, and the effective zone of UV-inactivation of colcemid due to short wavelength diffusion materialized by dashed line circles. Green arrows indicate MT nucleation in the vicinity of the nuclear envelope. Pink arrows indicate MT nucleation in the cytoplasm. (**e**–**h**) Close-ups of the oocyte shown in **a**–**d**. A: Anterior, P: Posterior; scale bar, 10 μm. (**i**–**l**) Selected frames extracted from a time-lapse movie, showing MT nucleation highlighted by EB1-GFP, before (**i**,**j**) and after UV-mediated colcemid inactivation (**k**,**l**). (**j**–**l**) are close-ups of the region boxed in **i**. The green arrow indicates MT nucleation at the nuclear envelope. Scale bar, 10 μm. (**m**–**p**) Quantification of the MT nucleation sites highlighted by EB1-GFP around the nuclear envelope. (**m**) MT nucleation sites highlighted by EB1-GFP are visualized as asters (red arrows) in the presence of colcemid that prevents further MT polymerization. (**n**) For each confocal section, the nuclear envelope is subdivided into anterior and posterior hemispheres, highlighted in orange and magenta respectively. (**o**,**p**) Quantitative measurement of MT nucleation sites. (**o**) 3D Schematic illustration of peri-nuclear asters of MT nucleation represented as green dots. (**p**) Bar plot presenting the total fluorescence intensity of anterior (orange) and posterior (magenta) EB1-GFP-positive peri-nuclear asters. (Intensities were summed up over 23 nuclei).

oocyte nuclear envelope. Previous studies had suggested that the MT minus ends-directed motors Dynein or the plus ends-directed motors Kinesin 1 may not be required for the nuclear migration[13,16,19]. However, these conclusions were based on fixed tissue experiments and in the light of our results require to be assessed with live-cell imaging approaches. Moreover, current tools make satisfactory Dynein inactivation at the onset of nuclear migration very difficult, given its crucial roles in oocyte development before stage 6. In other model systems, Kinesin-1 and Kinesin-3 have been implicated in moving the nucleus[4,39], and we cannot exclude a possible redundant activity of the

members of these kinesin families in the oocyte. Hence, the potential involvement of these MT-associated motors will need to be assessed.

Three-dimensional live imaging reveals that nuclear displacement comprises three phases: premigration, migration proper, and final cortical docking. Following the setting in motion of the nucleus, its movement towards the antero-dorsal cortex is a continuous and forward displacement. We have identified three distinct trajectories, either strictly cytoplasmic (STAD) or much more frequently two broad categories, APM and LPM, according to the site of the first contact with the plasma membrane.

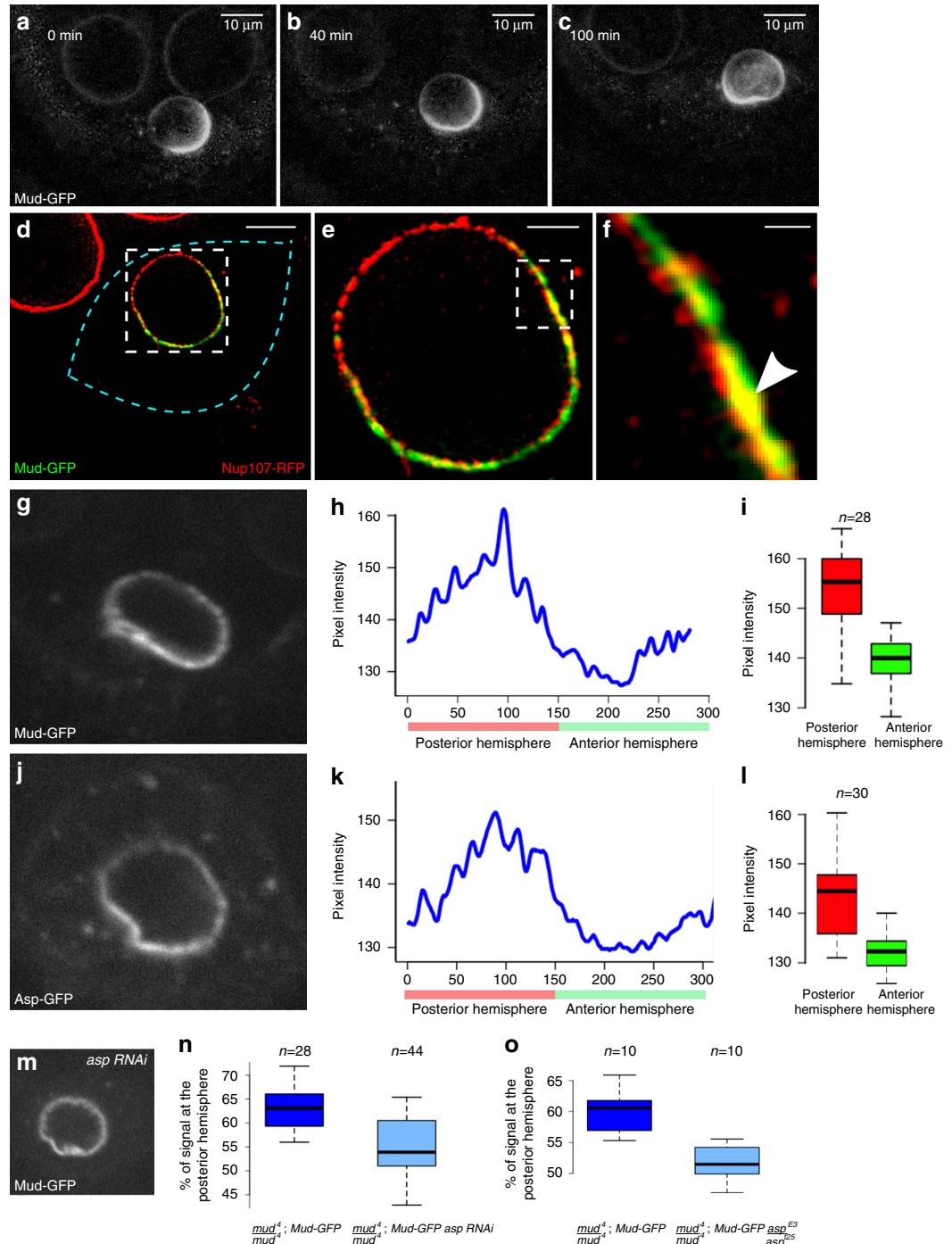

**Figure 6 | Asp controls Mud asymmetric distribution at the nuclear envelope.** (**a–c**) Selected frames extracted from a time-lapse movie showing Mud-GFP distribution during nuclear migration. (**d–f**) Live SIM microscopy illustrating the asymmetric distribution of Mud-GFP (green) at the nuclear envelope decorated by the nucleoporin Nup107-RFP (red). (**d**) The blue dashed line highlights the limits of a stage 6 oocyte. Scale bar, 10 μm. (**e**) Enlarged view of the region boxed in **d**; scale bar 2 μm. (**f**) Enlarged view of the region boxed in **e**; scale bar, 500 nm. The white arrowhead points to the co-localization of Mud-GFP with Nup107-RFP at the nuclear envelope. (**g–l**) Analysis of Mud-GFP (**g–i**) and Asp-GFP (**j–l**) distributions at the nuclear envelope of a stage 6 oocyte. (**h,k**) Intensity profiles of Mud-GFP (**h**) and Asp-GFP (**k**) along the nuclear envelope of the nuclei shown in **g,j**, respectively. The values 0–150 (red box) correspond to the posterior hemisphere. The values 150–300 (green box) correspond to the anterior hemisphere. (**i,l**) Box plot representations of Mud-GFP (**i**) and Asp-GFP (**l**) intensities in the posterior and anterior hemispheres (Mud-GFP: $n = 28$ – pValue $= 3.9 \times 10^{-6}$; Asp-GFP: $n = 30$ – pValue $= 3.3 \times 10^{-7}$). (**m**) Representative example of Mud-GFP distribution in a stage 6 *asp*-depleted oocyte. (**n,o**) Box plot representation of Mud-GFP intensity percentage in the posterior hemispheres of wild-type ($n = 28$) and *asp*-depleted oocytes ($n = 44$); (pValue $= 7.1 \times 10^{-8}$) (**n**) and of *wild-type* ($n = 10$) and *asp* mutant oocytes ($n = 10$); (pValue $= 2.2 \times 10^{-5}$) (**o**). In each case, the limits of boxplot correspond to the first and third quartiles, and the band inside is the median. The ends of the whiskers represent the minimal and the maximal values. Statistical differences have been assessed by a Wilcoxon–Mann–Whitney test.

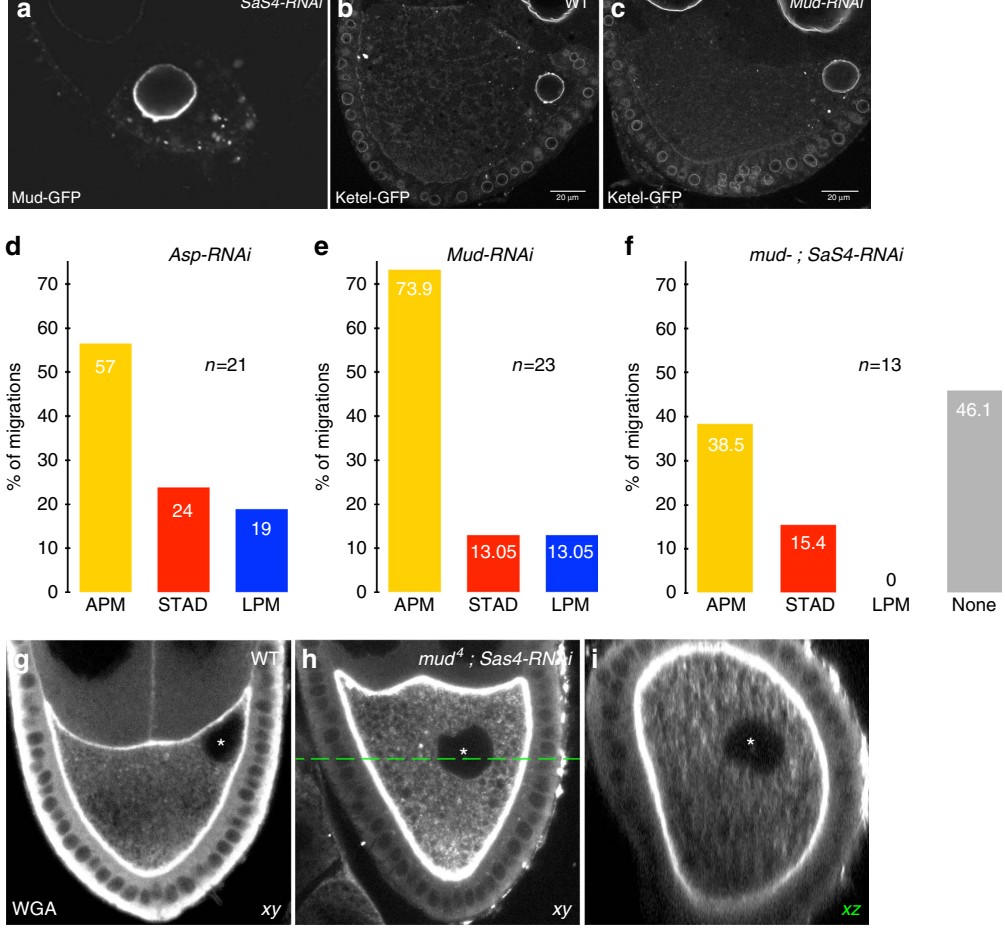

**Figure 7 | Mud and the centrosomes fulfil complementary functions in nuclear migration.** (**a**) Mud-GFP distribution is unaffected in stage 6 *Sas4*-depleted oocytes. (**b,c**) Antero-dorsal position of the nucleus revealed by Fs(2)Ket-GFP, in stage 9 wild-type (**b**) or *mud*-depleted (**c**) oocytes. (**d**) Percentages of the various migration paths taken by nuclei in *Asp*-RNAi oocytes ($n = 21$ independent nuclear migrations). (**e**) Percentages of the various migration paths taken by nuclei in *Mud*-RNAi oocytes ($n = 23$ independent nuclear migrations). (**f**) Percentages of the various migration paths taken by nuclei in *mud4, Sas4*-RNAi mutant oocytes ($n = 13$ independent nuclear migrations). Migration is abolished in 46% of the cases. (**g-i**) Stage 9 oocytes stained with WGA highlighting membranes. The nucleus is visualized by an absence of staining and marked with an asterisk. (**g**) wild type (**h**) *mud* (ref. 4); *Sas4sRNAi*. The green dashed line indicates the *xz* section displayed in **i** which illustrates the total lack of nuclear contact with any plasma membrane. WGA, wheat germ agglutinin

The APM and LPM trajectories are of equal duration and can be broken down into two consecutive periods. The first period that involves a displacement through the oocyte cytoplasm, starts at the onset of nucleus departure and ends up with its contact with the oocyte cortex. During the ensuing period, the nucleus follows the anterior (APM) or lateral (LPM) cortex before it reaches its final destination. This shows for the first time that migration of the oocyte nucleus is a two-step course.

Among the candidate components of the molecular machinery associated with MTs nucleated from the nuclear envelope, we identified Mud, a MAP that is specifically enriched on the posterior hemisphere of the envelope. Its localization to the nuclear membrane is independent of the centrosome. The association of Mud with the nuclear envelope is further supported by its interaction with the β–importin Ketel, which is localized around the nuclear envelope and has been shown to interact with nucleopore-associated proteins[22,40]. As a whole, this suggests that Mud interacts directly with the nuclear envelope. Its asymmetric enrichment on the nuclear envelope posterior hemisphere requires Asp. Importantly, Mud also has the ability to bind to MT minus ends and molecular motors[41]. Furthermore, Mud can stimulate MT formation *in vitro*[37], therefore suggesting that this MAP participates in MT nucleation at the nuclear envelope. Genetic ablation of Mud, which leads to both a decrease in nuclear velocity and migrations along the LPM, demonstrates that Mud contributes to nuclear displacement. Hence, in the oocyte, Mud may control the asymmetric MT assembly at the nuclear envelope and thereby promote the formation of MT subsets necessary for migration toward the LPM.

Our results reveal that the migration proceeds in two phases with a displacement through the cytoplasm followed by a sliding along the APM or the LPM. The initial commitment of the nucleus into the alternative APM or LPM routes appears to take place with equal probability. Nonetheless, our results indicate that distinct and genetically separable contributions of MT networks nucleated from either the centrosomes or the nuclear envelope favour the APM and LPM trajectories respectively. Because the migration speed is significantly decreased when either Mud or the centrosomes are ablated, this indicates that together they provide additive forces that contribute in a complementary manner to adjusting the migration speed to an optimum. We do not know how the nucleus slides along the APM or the LPM in the second

phase of its migration. However, one can speculate that pushing forces from the MTs should contribute and that nuclear association to the plasma membrane might also be an important factor. Indeed, once the nucleus hits the membrane, it does not appear to detach from it later on. Interestingly, several components of the plasma membrane and its associated cortex are required for proper nuclear positioning in the oocyte. The PAR proteins PAR-3 and PAR-1, as well as the phosphoinositide PIP2, are necessary for nuclear anchorage to the plasma membrane[23,42].

Our results indicate that centrosomes and perinuclear MAPs contribute to certain aspects of the architecture of the MT network, which is needed to accurately position the nucleus. Accordingly, it provides a conceptual framework for the study of cases where the nuclear envelope is particularly important for MT assembly and nuclear positioning. This may be especially relevant for muscle development and associated diseases such as centro-nuclear myopathies, in which a defective nuclear envelope impairs nuclear migration[1].

## Methods

**Fly strains.** *Drosophila* stocks and crosses were kept under standard conditions at 25 °C. The following stocks were used: Fs(2)Ket-GFP (P{PTT-un1}Fs(2)Ket$^{GFP}$) (ref. 43), ubi-PH$^{PLC-\delta1}$-RFP (ref. 24), Jupiter-GFP (ref. 25), UASp-EB1-GFP (ref. 25), Nup107-RFP (ref. 22), Asl-tdTomato (ref. 44), cnn-GFP (ref. 28), Spd-2-GFP (ref. 28), LamC-GFP (LamC$^{CB04957}$) (ref. 45), mud (ref. 4), Df(1)KA9 (Bloomington Drosophila Stock Center), GFP:Mud transgenes under the control of mud endogenous promoter[46], ubi-asp-GFP (refs 38,47), asp$^{E3}$ (ref. 47), asp$^{t25}$ (ref. 38). The nos-GAL4 driver stock P{GAL4::VP16-nos.UTR}CG6325$^{MVD1}$ and the shRNA (TRiP, Harvard Medical School) stocks asl$^{HMS01453}$, Sas-4$^{HMS01463}$, mud$^{HMS01458}$ and *asp*$^{GL00108}$ (Bloomington Drosophila Stock Center) were used for RNAi experiments.

**Control for the efficiency of shRNAs.** Mud is essential for spindle assembly during meiosis II. Mud-depleted females are therefore sterile[48]. Homozygous mutant females for *mud* lay eggs that fail to develop and to hatch. *Mud* RNAi efficiency was monitored on the basis of female sterility. We observed that 95% of the eggs failed to hatch ($n = 300$). Furthermore, *GFP:Mud* expression was impaired by mud shRNA. Sas4 and Asl are essential for centriole formation and centrosome biogenesis[27,49]. The expression during oogenesis of such factors is essential for centriole replication during early embryogenesis[30]. In their absence, embryos fail to develop and to hatch. Sas-4 and asl RNAi efficiency were monitored by evaluating female sterility. For Sas-4 ($n = 300$) and asl ($n = 400$), 100% of the eggs failed to hatch. Asp is required to cross-link spindle MTs. In the absence of Asp, spindles remain unfocused and Asp mutant females are sterile. The efficiency of Asp RNAi-mediated knockdown was monitored by female sterility. 100% of the eggs failed to hatch ($n = 280$).

**Immunohistochemistry.** Ovaries were hand dissected and fixed in PBS with 4% paraformaldehyde prior over night incubation with wheat germ agglutinin in PBS with 0,1% tween. Wheat germ agglutinin was used at 1/100 (Molecular Probes).

**Live imaging and nuclear migration tracking.** Egg chambers were cultured in a 150 µm deep micro chamber filled with Schneider medium supplemented with insulin, FBS, penicillin and streptomycin according to ref. 50 and sealed on one side with a 0.17 µm coverslip matching the characteristics of the facing objective lens, and a membrane permeable to oxygen on the other side. Imaging was carried out with either a Leica DMIRB microscope coupled to a spinning disk module (Yokogawa CSU10) and a charge-coupled device (CCD) camera (Coolsnap HQ2, Photometrics) (488, 561 lasers and a × 40 objective (HCX PL APO, 1.25NA, Leica)), or a Zeiss Axio Observer LSM780 microscope with GaAsP detectors (488, 561 lasers, × 63 objective Plan Apo, 1.4NA, oil immersion, Zeiss). Images were acquired with time-steps ranging from 2 s to 20 min for up to 10 h. Time-lapse recordings were processed with Fiji. For each frame at each time point, the position of the oocyte was corrected for rotations around z axis and translations of the egg chamber during the acquisition period. The geometric centre of the nucleus was then tracked using the MtrackJ plugin throughout migration.

**Analysis of nuclear migratory paths.** Twenty-four independent nuclear migrations were analysed. To characterize the different migration events, the first contact point of the nucleus with the plasma membrane was taken into account to distinguish two categories, one (22 events) where the nucleus hit the plasma membrane before riding along it until the end of migration, and the

other (two events) where the nucleus migrated straight through the cytoplasm (STAD) to the antero-dorsal cortex. To further analyse the 22 events corresponding to the first category, the distribution of the first contact point along the APM and the LPM was investigated. A contact point was defined when the distance between the nuclear envelope (labelled with Fs(2)Ket-GFP) and the plasma membrane (labelled with PH$^{PLC-\delta1}$-RFP) were shorter than 1 µm. Images were acquired orthogonally to the optical axis of the microscope (z axis) (Supplementary Fig. 1A).

To compare the initial contact points corresponding to the different oocytes analysed, we have placed all these points on a single reference oocyte. To position an initial contact point onto this reference oocyte, we considered its position with respect to some reference points as follow. The position of the contact point (point 1) in the acquisition plane (2,3,4,5) (Supplementary Fig. 3A,B) was reconsidered in the only equatorial plane (2′,3″,4′,5″) that contains this contact point (Supplementary Fig. 3A,C). Then the position of the contact point was normalized with respect to points 3″ and 5″ of the equatorial plane, in order to compare the different initial contact points in all samples. The coordinates of the points 1, 2, 2′, 3, 3′, 4, 4′, 5 and 5′ were deduced from the acquisition planes (B) considering that the oocyte exhibits an axial symmetry around the AP axis. Since the distance d(2′, 3′) is equal to the distance d(2′,3″) (Supplementary Fig. 3A) due to axial symmetry, the ratio 'r' is used to set the position of the initial contact point in the reference oocyte, r = d(2′,1)/d(2′,3″) (Supplementary Fig. 3C). A reference oocyte was defined by three points A, P, D (for anterior, posterior and dorsal), where A and P correspond to 2′and 4′, while D corresponds either to 3″ or 5″. For APM migrations, point 1 was then positioned on the diagram according to the relation: d(A, 1) = r × d(A, D) (Supplementary Fig. 3D). For LPM migrations, the relation was d(P, 1) = r × d(P, D), similarly. Initial contact points were then positioned on the reference oocyte for the 22 migrations analysed (Supplementary Fig. 3E). Since the oocyte presents an axial symmetry along the AP axis, all 22 nuclear initial contact points were displayed on a reference hemi-oocyte (Supplementary Fig. 3F).

**Analysis of MT distribution in nuclear migration.** MT distribution was analysed in 31 independent nuclear migrations using the MAP Jupiter tagged with GFP[51]. All the migrations have been normalized towards the right. For each time point, an image was obtained from the average projection of three confocal sections (step size 2 µm) encompassing the equatorial plane of the nucleus (Fig. 1f,g). The oocyte was divided in 12 sectors of 30° and further subdivided into trapezoid boxes representing a fixed volume (Fig. 1h). For each time point, the mean signal intensity was calculated within each box (Fig. 1i) and displayed in a graphical representation of the oocyte (Fig. 1j). Mean values for the 31 independent migrations were similarly obtained for three representative time points (Fig. 1k–m).

**Laser-mediated ablation.** Laser ablations were performed on an inverted scanning confocal microscope (TCS SP2 AOBS MP, Leica Microsystems) coupled to an 80 MHz laser head (MaiTai DeepSee, Spectra-Physics) delivering infrared femtosecond pulses. An acousto-optic modulator was used to adjust the power from 2% to 75% of the laser output power. To perform local ablation, the pulsed laser beam was focused on the region of interest (ROI) with a microscope objective (1.25NA, × 40 oil immersion, HCX PL APO, Leica), which transmits 85% of the incoming near-infrared light. The focused beam was set to 860 nm and targeted onto different cytoplasmic areas for 40 ms with an average power of 330 mW measured at the entrance of the scanning head. The ablation volume had an approximate 500 nm diameter and 1 µm depth. For long-term imaging, laser-mediated ablations were carried out on the above-described system, while subsequent imaging for 10 h was acquired on a Leica spinning disk confocal microscope in order to minimize phototoxicity. Laser ablations followed by short-term imaging were carried out with a ultraviolet-laser (355 nm, 400 ps, 20 kHz) coupled to a Nikon Eclipse Ti2 microscope associated with a spinning disk module (Yokogawa CSU X1) and a CCD camera (Coolsnap HQ2, Photometrics). A × 60 Nikon objective (1.4 NA oil immersion) was used.

**Super resolution structured-illumination imaging.** Super-resolution structured-illumination microscopy images were obtained using a Zeiss Elyra PS.1 microscope (488 and 561 nm excitation wavelengths) with a × 63 Zeiss objective (Plan Apo 1.4NA oil immersion). The fluorescence signal was detected with an EMCCD camera (iXon-885, Andor). Centriolar proteins co-localization was assessed on egg chambers fixed with 4% PFA and mounted in Citifluor medium. Co-localization of the MUD and NUP107 proteins on the nuclear envelope was assessed *ex vivo*, in live egg chambers mounted in a Voltalef 10 s oil droplet. 0.17 mm high-performance Zeiss coverslides were used. Images were acquired for up to 25 µm in depth.

**Link between indentation force and nuclear displacement.** To determine the vector of the force(s) associated with the nuclear indentation(s), the envelope was detoured (black dotted line) in its plane of largest area (equatorial plane), and its geometric centre N$_1$ was determined (Supplementary Fig. 1B). A circular model envelope (red dotted circle) with a radius ($R_{model}$) equal to the average one observed in the biological sample (black dotted line) was then superimposed. Indentations were defined as the contiguous portions of the detoured envelope (black) with a radius smaller than the circular model envelope (red)

($d_1$, ..., $d_i$ < $R_{model}$ with $d_i$ being the distance between point i and $N_1$). For all nuclear indentations, the average $d_i$ reflecting the depth of each indentation was calculated. As a first approximation, it is proportional to the magnitude of the force ($\|\mathbf{F}\|_{indentation}$) that creates the indentation. The direction of this force was calculated as orthogonal to the mean tangent to the corresponding indentation. The resultant of the forces that create n indentations is $\mathbf{F} = \mathbf{F}_{indentation 1} + \mathbf{F}_{indentation 2} + \mathbf{F}_{indentation n}$ (Supplementary Fig. 1B for an example of a nucleus displaying two indentations (green and blue sections)). During migration, $N_1$ and $N_2$ were defined as nuclear positions at two consecutive time points $t_1$ and $t_2$. The angle $\alpha$ between the resulting indentation force $\mathbf{F}$ and the vector $\mathbf{N1N2}$ of the observed displacement was then calculated. Forces that create indentations with $\alpha \geq 90°$ cannot support nuclear displacement. Only forces associated with $\alpha < 90°$ can sustain the observed motility. In total, 429 $\alpha$ angles were measured by repeating the same calculation between all pairs of consecutive time points, for 50 migrating nuclei (Supplementary Fig. 1C).

**Nuclear correlation between displacement and indentation.** The cross-correlation between a normalized displacement and the angle between the indentation force and the displacement vector for the nucleus was evaluated for each movie. The possibility of a delay for the correlation between the displacement and the angle between the indentation force and the displacement vector was also considered to detect a putative viscous component in particular. The cross correlation was evaluated considering a potential lag time of up to 75 min. The correlation analysis was performed with the respective lag-time in min: 0, 15, 30, 45, 60, 75. The different correlation functions are the following: 0 min: $y = -0.0006x + 0.5564$, 15 min: $y = -0.0008x + 0.5584$, 30 min: $y = -0.0001x + 0.484$, 45 min: $y = -4 \cdot 10^{-7}x + 0.4698$, 60 min: $y = -0.001x + 0.3915$, 75 min: $y = -0.0026x + 0.2462$. The theoretical function is represented with an $R^2$ of $-1$.

**MT depolymerization by colcemid.** Dissected egg chambers were cultured in the conditions described in 'Live imaging and nuclear migration tracking', except for the addition of colcemid (0.25 µg ml$^{-1}$) to the culture medium. Images were acquired with a Leica DMIRB microscope coupled to a spinning disk module (Yokogawa CSU10) with a ×40 objective (1,25NA HCX PLAPO Leica) and a CCD camera (Coolsnap HQ2, Photometrics).

**Colcemid inactivation.** Egg chambers were first incubated under agitation during 1 h in Schneider medium supplemented with insulin, FBS, penicillin, streptomycin according to ref. 50 and colcemid (0.25 µg ml$^{-1}$). Chambers were then mounted directly in a droplet of Voltalef 10 s oil onto a 0.17 mm cover slips. Images were acquired with a Nikon microscope (Eclipse Ti2) coupled to a spinning disk module (Yokogawa CSU X1), a CCD camera (Coolsnap HQ2, Photometrics), and a 488 nm laser to excite GFP. Local ultraviolet-mediated inactivation of colcemid in a defined ROI (Fig. 5a,b) was carried out with a pulsed ultraviolet-laser (355 nm, 400 ps, 20 kHz). Global ultraviolet-mediated inactivation (Fig. 5c) was carried out with fluorescence HBO lamp.

**Perinuclear asters quantification.** EB1-GFP; Asl-tdTomato egg chambers were prepared following the 'Colcemid inactivation' protocol, skipping the ultraviolet inactivation step. The fluorescent signal of peri-nuclear asters (positive for EB1-GFP but negative for Asl-tdTomato) was quantified on stacks of sections encompassing the whole nucleus, along with the z axis. Noise subtraction was performed with subtracting background (Fiji), based on the rolling ball algorithm. For each confocal section, the nuclear envelope circumference was divided into anterior (A) and posterior (P) halves on which total EB1-GFP signal was quantified with polyline and profile tools (Fiji). The mean value of the circumference signal in each confocal section was then subtracted in order to minimize the noise. The ratio (P/A) between the two nuclear hemispheres was calculated from the sum of values obtained in each half of all nuclear sections.

**Data availability.** We declare that all data supporting the findings of this study are available within the article and its Supplementary Information files or from the corresponding author upon reasonable request.

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

## Acknowledgements

We thank the team of the ImagoSeine facility at the Institut Jacques Monod for their help and support, the France-Bio-Imaging national research structure (ANR-10-INBS-04), Renata Basto, Maria Rujano, Delphine Gogendeau, Nasser Rusan, the Bloomington Drosophila Stock Center and the Developmental Studies Hybridoma Bank for fly stocks and reagents. We are grateful to Nicolas Borghi, Véronique Brodu, Julien Dumont, Nicolas Minc, Lionel Pintard for critical comments on the manuscript. We are grateful to the laboratory members for helpful discussions. We thank Laetitia Besse, Edouard Jaumouillé, Mathilde Lediuzet, Jean-Yannick Petit, Fabrice Licata for technical assistance, and Thomas Rubin for help with the colcemid experiments. NT is a fellow of France-Bio-Imaging. KL is a fellow of the 'Ligue Nationale Contre le Cancer' (JB/GB/MA/IQ-9823 and GB/MA/IQ-10594). This work was supported by the CNRS, by the Association pour la Recherche sur le Cancer (grant SL220100601358), (grant PJA 20141201756), (grant PJA 20161204931) and by the 'Ligue Nationale Contre le Cancer' (grant RS11/75-34).

## Author contributions

N.T., J.A.L., Fr.B., K.L. and A.G. designed the experiments, interpreted the results and wrote the manuscript. N.T., J.A.L., Fr.B., K.L. carried out the experiments. Fl.B., C.M., Y.B. generated Mud-GFP flies, O.F. performed SIM microscopy experiments. Fl.B., Y.B., M.C. corrected the manuscript.

## Additional information

**Competing interests:** The authors declare no competing financial interests.

