## [Peer Review File · Nature Communications]

Reviewers' Comments:

Reviewer #1 (Remarks to the Author)

Tissot et al submitted a revised version of "Complementary molecular cues ensure a robust microtubule-dependent nuclear positioning in the *Drosophila* oocyte", in which they greatly increased the robustness and clarity of their analysis. I now strongly believe their conclusions are entirely supported and would recommend the paper be accepted more or less as is. I have two minor comments.

1. I didn't say this quite clearly enough in my initial review, but I don't think, given the importance of nuclear deformations to their modification of the hypothesis in Zhao, et al, that there is enough quantification of nuclear shape. Nuclear shape can change for many reasons and I would have no expectation the authors would account for every one of them. But what kind of deformations are we talking about? Some of the images from the manuscript really look like these deformations are so small that one wonders how significant they are. And is there any correlation between the angle of movement and the magnitude of deformation? I am asking the authors to slightly expand on the the analysis in Fig. S1B and C. What are the magnitudes of deformation across the measurements? Is there a stronger correlation with displacement when this is taken into account? It is not only angle between Force and Displacement vectors that is important, it is also the projection of F (as deduced from deformations) along D. This would be a good measurement. I also don't understand why the authors don't use correlation coefficients in the actual manuscript, as suggested by myself and another reviewer. As it stands, Fig 1E is probably good enough, but it seems like it would strengthen the analysis, and make it less arbitrary, if a proper correlation analysis were done here.

2. Very minor comment: end of the last full paragraph of p. 12 - "Hence Asp controls Mud asymmetric distribution..." should be "Hence Asp is correlated with Mud....", no?

Anyway, it is looking like a strong manuscript now. Besides the two above suggestions I am satisfied that their revision completely addresses the reviewer comments and their conclusions look justified now.

Reviewer #2 (Remarks to the Author)

Reviewer #3 (Remarks to the Author)

My previous concerns have been fully addressed in the revised version of the manuscript.

Reviewer #4 (Remarks to the Author)

This paper addresses the mechanism of microtubule-dependent nuclear migration in *Drosophila* oocyte. A previous study (Zhao et al., Science 2012) proposed a model, in which growing microtubules nucleated from the centrosomes behind the nucleus push the nucleus forward. The major argument supporting this mechanism was the observation of nuclear indentation at the side which is being pushed. In the current work, the authors show that a more detailed 3D analysis of

nuclear movement in this system fails to detect a correlation between the position of the indentation and the direction of nuclear movement. This invalidates the previous model and by itself deserves publication.

The other parts of the paper, reporting photoablation experiments, as well as depletion of centrioles and Mud/NuMA and Asp, are interesting but seem overinterpreted. In the opinion of this reviewer, this does not need to preclude publication, but the conclusions of the paper need to be toned down and accompanied by a more balanced discussion.

First, the logic behind the interpretation of the photoablation experiments is difficult to understand – surely, any local microtubule photoablation will lead to their rapid but potentially disorganized regrowth on the scale of seconds or minutes? It is thus unclear how any inferences can be made from such experiments on the forces acting at longer time scales. The observation that ablation at the posterior has a larger impact might simply reflect the fact that there are more microtubules on that side or that centrioles are ablated.

For centriole depletion, Zhao et al showed that this may lead to formation of acentrosomal MTOCs, a point that should be discussed in the paper. As to the involvement of Mud and Asp, these observations are very interesting but since it is not really clear how the microtubule system is organized in oocytes with or without these proteins, and how microtubules nucleated from the nuclear envelope might contribute to nuclear positioning, it seems premature to talk about “two independent complementary contributions from the MTs”. It seems more likely that centrosomes and MAPs contribute to certain aspects of the architecture of the microtubule network, which is needed to position the nucleus. In the opinion of this reviewer, completely ignoring potential motor contributions also seems problematic: dynein is so important for different aspects of microtubule and cell architecture that ruling out its involvement is extremely challenging. As to kinesins, in the mammalian systems, both kinesin-1 and kinesin-3 have been implicated in moving the nucleus, and it is not clear whether the possibly redundant activities of the members of these kinesin families have been tested in fly oocytes.

Reviewer 1

1: Concerning the nuclear deformations

We indeed observed a variation for the nuclear deformations. We observed cases with several or only weak deformations of the nuclear envelope as indicated in the result section p. 6. For this reason, we have considered for each case a vector resulting from the various nuclear deformations as illustrated in the supplemental Fig. 1B. We would like to further emphasize that the originality of our study is to analyse the global forces applied to the nuclear envelope by calculating a vector of force resulting from all the deviations of the nucleus shape from a perfect circle.

2: Concerning the correlation analysis between the indentation of the nucleus and its displacement

We would like to kindly remind that those results had been already described in the previous version of the manuscript according to the reviewer suggestion (see cover letter September 15, 2016 p 3).

We have evaluated the cross correlation between a normalized displacement and the angle between the indentation force and the displacement vector for the nucleus for each movie (Sup Fig. 1D). We observed no correlation with any angles as revealed by a correlation coefficient R^2 of 0.016 (Sup Fig. 1D). Furthermore, as also suggested by reviewer 1, we have also considered the possibility of a delay (i.e. a potential lag time of up to 75 min) in the calculation of the cross correlation between the displacement and the angle between the indentation force and the displacement vector (Sup Fig. 1E) in order to consider a putative viscous component of the cytoplasm.

Importantly, in order to improve the visibility of this cross correlation analysis, the sup fig 1E has now been included in the main figure as Fig. 1F. The result section is unchanged as the sup fig 1D and the new Fig.1 F were already mentioned in the manuscript Results section page 6 “ *This possibility was thus investigated by evaluating the cross-correlations between the distance covered by the nucleus and the angle between indentation force and displacement vector, with a lag of 0, 15, 30, 45, 60 and 75 min.*”.

3: Concerning a potential correlation between the angle of the movement and the magnitude of deformation.

According to the reviewer suggestion, we have analysed a potential correlation between two values. One is the measurement of the angle of the vector resulting from the various nuclear

indentations (supplemental Fig. 1B) and the displacement vector for the nucleus, and the second the strength of the major indentation (estimated by the ratio (r) of the indentation depth over the nucleus radius).

We have performed this analysis for 313 angles corresponding to 38 independent nuclear migrations. We have distinguished four categories for r : $r < 2.5\%$; $2.5\% < r < 5\%$; $5\% < r < 10\%$; $r > 10\%$ corresponding to a, very weak, weak, medium and strong deformation respectively. We observed that the strength of the nuclear deformation has very little, if any, influence on the correlation between the displacement and the nuclear indentations.

($r < 2.5\%$ $n = 18$; $2.5\% < r < 5\%$ $n = 174$; $5\% < r < 10\%$ $n = 65$; $r > 10\%$ $n = 56$; $n^{\text{total}} = 313$)

4: Concerning the comment about Asp controlling Mud asymmetric distribution. :

To clarify our conclusion, we have rephrased our conclusion at the end of the result section concerning Asp as follows: "*Hence Asp controls Mud asymmetric distribution at the nuclear envelope*". has been changed to: "*Hence, Asp is not necessary for mud localisation at the nuclear envelope but for its asymmetric distribution*".

Reviewer 4

We agree with the different points raised. We have toned down the conclusion concerning the points raised by the reviewer and we have rewritten the discussion taking into account all the suggestions of the referee.

1: Concerning the conclusion «Two independent complementary contributions from the MTs »

The terms complementary and independent have been removed.

The previous title: *Complementary molecular cues ensure a robust microtubule-dependent nuclear positioning in the Drosophila oocyte* has been changed to *Distinct molecular cues ensure a robust microtubule-dependent nuclear positioning in the Drosophila oocyte*.

Accordingly, in the introduction page 4: "We reveal that complementary contributions from the centrosomes and the NuMA homolog Mud at the nuclear envelope in a microcephaly protein Abnormal spindle-dependent mode, ensure the robustness of asymmetric nuclear positioning through alternative migratory routes" has been changed to: We reveal that the dual contributions from..."

In the last section of the results page 13:

The title: *Mud and the centrosomes fulfil complementary functions* has been changed to *Mud and the centrosomes confer robustness to nuclear migration*

We agree with the referee comments that it seems more likely that centrosomes and MAPs contribute to certain aspects of the architecture of the microtubule network, which is needed to position the nucleus. Therefore, at the end of the discussion page 17: "Altogether, our results lead to an unanticipated model for how the nucleus is accurately localised to the antero-dorsal cortex through a robust process involving two independent but complementary contributions from the MTs" has been changed to : *Our results indicate that centrosomes and perinuclear MAPs contribute to certain aspects of the architecture of the microtubule network, which is needed to accurately position the nucleus.*

2: Concerning the interpretation of the photo ablation

We have toned down the interpretation of the photo ablation, and accordingly, the title of the relevant section “*The nucleus is pushed by MTs*” page 7 has been modified to: *MT forces applied on the nucleus*.

We have also included a new paragraph in the discussion page 14 concerning this point where we emphasize the complexity of the interpretation of these photoablations experiments :

In an attempt to further decipher the forces involved in nucleus displacement, our laser ablation experiments showed that major forces involved are concentrated at the posterior of the oocyte and we interpreted this observation as an evidence of the nucleus being pushed, in accordance with a previous study (Zhao et al., 2012). However, further analyses of MT organization in Drosophila oocyte and especially the report of Mud and Asp, two previously shown MT minus-end associated proteins (Bowman et al., 2006; Schoborg et al., 2015), make the interpretation of these photo-manipulation experiments more complex.

3: Concerning the putative role of the MT motors.

The evidence suggesting that the motors do not play an important role, are still incomplete and leave open future investigations to address this point.

Nevertheless, we have inserted a new paragraph in the discussion, where we included the suggestion of the referee concerning a possible redundancy between kinesin 1 and kinesin 3.

Previous studies had suggested that the MT minus ends-directed motors Dynein or plus ends-directed motors Kinesin 1 may not be required for the nuclear migration, (Duncan and Warrior, 2002; Januschke et al., 2002; Zhao et al., 2012). However, these conclusions were based on fixed tissue experiments and in the light of our results require to be assessed with live-cell imaging approaches. Moreover, current tools make satisfactory Dynein inactivation at the onset of nuclear migration very difficult, given its crucial roles in oocyte development before stage 6. In other model systems, Kinesin-1 and Kinesin-3 have been implicated in moving the nucleus (Fridolfsson and Starr, 2010; Tsai et al., 2010), and we cannot exclude a possible redundant activity of the members of these kinesin families in the oocyte. Hence, we cannot rule out the potential involvement of these MT associated motors and further analysis will be required to address this question.

References

- Bowman, S.K., R.A. Neumuller, M. Novatchkova, Q. Du, and J.A. Knoblich. 2006. The *Drosophila* NuMA Homolog Mud regulates spindle orientation in asymmetric cell division. *Dev Cell*. 10:731-742.
- Duncan, J.E., and R. Warrior. 2002. The cytoplasmic dynein and kinesin motors have interdependent roles in patterning the *Drosophila* oocyte. *Curr Biol*. 12:1982-1991.
- Fridolfsson, H.N., and D.A. Starr. 2010. Kinesin-1 and dynein at the nuclear envelope mediate the bidirectional migrations of nuclei. *J Cell Biol*. 191:115-128.
- Januschke, J., L. Gervais, S. Dass, J.A. Kaltschmidt, H. Lopez-Schier, D. St Johnston, A.H. Brand, S. Roth, and A. Guichet. 2002. Polar transport in the *Drosophila* oocyte requires Dynein and Kinesin I cooperation. *Curr Biol*. 12:1971-1981.
- Schoborg, T., A.L. Zajac, C.J. Fagerstrom, R.X. Guillen, and N.M. Rusan. 2015. An Asp-CaM complex is required for centrosome-pole cohesion and centrosome inheritance in neural stem cells. *J Cell Biol*. 211:987-998.
- Tsai, J.W., W.N. Lian, S. Kemal, A.R. Kriegstein, and R.B. Vallee. 2010. Kinesin 3 and cytoplasmic dynein mediate interkinetic nuclear migration in neural stem cells. *Nat Neurosci*. 13:1463-1471.
- Zhao, T., O.S. Graham, A. Raposo, and D. St Johnston. 2012. Growing microtubules push the oocyte nucleus to polarize the *Drosophila* dorsal-ventral axis. *Science*. 336:999-1003.

Reviewers' Comments:

Reviewer #1 (Remarks to the Author)

The revisions have satisfied all of my suggestions/comments. I recommend acceptance at this point.

Reviewer #4 (Remarks to the Author)

The revisions improved the paper and I support its publication, provided that the authors make the following minor textual change: while discussing the results of photoablation shown in Figure 2, the authors must explicitly state already in the Results that microtubule regrowth after photoablation would occur on a much shorter time scale than the studied effects on nuclear positioning. This is essential, because not all readers will be aware of this important caveat of this experiment.

Reply to reviewer's comments

Reviewer 4

We agree with the point raised.

While discussing the results of photoablation shown in Figure 2, the authors must explicitly state already in the Results that microtubule regrowth after photoablation would occur on a much shorter time scale than the studied effects on nuclear positioning.

Accordingly, in the result section concerning the Figure 2 page 8, we have included the sentence proposed by the reviewer: “It should be noted that in such case, MT regrowth after photoablation would occur on a much shorter time scale than the studied effects on nuclear positioning.”